# Association between Non-Typhoidal Salmonella Infection and Growth in Children under 5 Years of Age: Analyzing Data from the Global Enteric Multicenter Study

**DOI:** 10.3390/nu13020392

**Published:** 2021-01-28

**Authors:** Rina Das, Md. Ahshanul Haque, Mohammod Jobayer Chisti, Abu Sayed Golam Faruque, Tahmeed Ahmed

**Affiliations:** 1Nutrition and Clinical Services Division, icddr,b, Dhaka 1212, Bangladesh; rina.das@icddrb.org (R.D.); ahshanul.haque@icddrb.org (M.A.H.); chisti@icddrb.org (M.J.C.); tahmeed@icddrb.org (T.A.); 2James P. Grant School of Public Health, BRAC University, Dhaka 1212, Bangladesh; 3Department of Global Health, University of Washington, Seattle, WA 98104, USA

**Keywords:** non- typhoidal *Salmonella*, GEMS, under 5 children, z-score, undernutrition, diarrhoea

## Abstract

Non-typhoidal *Salmonella* (NTS) is one of the less focused on infections and is often associated with faulty child nutrition in the developing world. This study aimed to evaluate the association of NTS infection with growth faltering among children under the age of five. We analyzed data from 378 fecal NTS positive children with both moderate-to-severe diarrhea (MSD) and asymptomatic infection from the seven countries of South Asia and sub-Saharan Africa during enrolment and on day 60 follow up in the Global Enteric Multicenter Study (GEMS) for the period of December 2007 to March 2011. Children not associated with fecal NTS (*n* = 1134) were randomly selected from the same dataset (1:3 ratio) as a comparison group. The association between an explanatory variable and the outcome variable was longitudinally tested using generalized estimating equations (GEE), where the dependent variables were height-for-age (HAZ), weight-for-age (WAZ), and weight-for-height (WHZ) z-score, and the independent variable was the presence of fecal NTS. The GEE multivariable model identified a negative association between fecal NTS and WAZ (coefficient: −0.19; 95% CI (confidence interval): −0.33, −0.04, and *p* value = 0.010), WHZ (coef: −0.19; 95% CI: −0.34, −0.05, and *p* value = 0.007), and HAZ (coef: −0.13; 95% CI: −0.27, −0.01, and *p* value = 0.073) after adjusting for age, gender, diarrhea, breastfeeding status, mothers’ education, number of children under the age of five, household size by the number of people regularly sleep at the home, handwashing practice, source of drinking water, wealth index, presence of co-pathogens, comorbidity, and study sites. In the GEMS, where children were followed during 50–90 days of enrolment, the presence of fecal NTS harmed the child’s anthropometric outcomes. Minimizing potential exposure to NTS is needed to curb worsening child undernutrition.

## 1. Introduction

An approximate 3.4 million global invasive enteritis cases per year are caused by non-typhoidal *Salmonella* (NTS) infection [1]. NTS is a gram-negative bacterium responsible for illness in both humans and animals worldwide. It causes self-limiting illness, which is usually asymptomatic. In the case of acutely symptomatic individuals present with diarrhea, NTS can affect nutritional status due to anorexia causing reduced intake, less absorption, and excess energy expenditure. It is responsible for an estimated 150,000 human deaths worldwide annually [2]. The Institute of Health Metrics and Evaluation reported that 81,300 diarrheal deaths were associated with enteric NTS disease in 2010 [3,4]. NTS species are most commonly present in animals like poultry, farm animals, reptiles, and pets [5].

Evidence-based information is lacking in South Asia about the epidemiology of NTS infection. It is a common cause of bacteremia in children and adults in sub-Saharan Africa, especially in areas with increased rates of human immunodeficiency virus (HIV) infections and malaria [6]. Most South Asian studies have concentrated on monitoring antimicrobial resistance trends [7,8] and circulating serotypes [9]. Despite limited knowledge, increasing NTS infection rates have been reported, particularly among HIV-infected populations [10,11].

Chronic fecal carriage of NTS species is rare and occurs less frequently than does chronic fecal carriage of *S. typhi.* But NTS more vigorously and durably colonize the intestinal tract than do typhoidal strains, thereby providing more prolonged opportunity for spreading. Such spread might also be enhanced by intestinal inflammation resulting from chronic diarrheal disease, parasitic infection, or suboptimal nutrition [12].

Gastrointestinal disorders and malnutrition have an undeniable link. Diarrhea is one of the most prevalent gastrointestinal disorders that can affect nutritional status [13]. The diversity in nutritional consequences in diarrheal illness has been reported to be etiology specific. Due to the low prevalence of the disease, limited up-to-date evidence exists on NTS infection and malnutrition. Diarrhea and NTS infection impacting on child growth has been reported by some researchers [14,15,16]. However, a study has become imperative to demonstrate conclusive evidence suggesting the role of NTS infection following diarrhea on child growth.

This study was conducted to examine the relationship between NTS infection and growth in resource-limited settings of sub-Saharan Africa and South Asia among diverse children under the age of five.

## 2. Materials and Methods

### 2.1. Study Site

The study was conducted at 4 African sites (The Gambia, Kenya, Mali, and Mozambique) and 3 Asian sites (Bangladesh, India, and Pakistan). While selecting these seven field sites main consideration was moderate-to-high under-5 child mortality in Africa and Asia. Details about the study sites were described elsewhere [17,18].

### 2.2. Study Design and Study Participants 

The design and methodology of the Global Enteric Multicenter Study (GEMS) were mentioned earlier [18]. For this study, briefly, data were extracted from cases and controls enrolled in the GEMS, a three-year (December 2007 to March 2011), prospective, age-stratified, matched case-control study of moderate-to-severe diarrhea (MSD) episodes [18] among children <5 years of age, residing within a specified and enumerated population [18,19]. Within 14 days of each enrolled case, 1–3 randomly selected age- and sex-matched community children were enrolled as controls from the same or neighborhood communities [20]. The study had a well-defined standardized recruitment protocol [18]. The published [19], working hypothesis [17], epidemiology [18], clinical [21], laboratory [18], and statistical methods [22] of GEMS have been described elsewhere [23]. In our study, children having NTS in stool constituted cases (*n* = 378) and site-specific randomly selected children with no NTS in stool constituted controls (*n* = 1134) (Figure 1). For the present analysis, controls were randomly selected country-wise from the database of enrolled GEMS children by computer-generated randomization using SPSS v. 25.0 (SPSS Inc., Chicago, IL, USA). To increase the statistical power of our analyses, we used an unmatched case-control ratio of 1:3.

### 2.3. Stool Sample Collection and Fecal Microbiology

Stool specimens for the GEMS were examined concomitantly from December 2007 to March 2011 for every child at the time of enrolment. Stool samples were processed using the GEMS laboratory procedure protocol [24,25]. A comprehensive etiologic study including NTS was undertaken. Bacterial pathogens (*Salmonella*, *Shigella*, *Vibrio cholerae*, *Campylobacter*, *Aeromonas* spp. and (enterotoxigenic, enteropathogenic, and enteroaggregative) *Escherichia coli*), viruses (rotavirus, norovirus, sapovirus, astrovirus, and adenovirus), and protozoa (*Giardia intestinalis*, *Entamoeba histolytica*, and *Cryptosporidium* spp.) have been identified by standard laboratory methods referred elsewhere [24]. NTS has been isolated using standard bacteriological methods and biochemically suspected NTS isolates [urea (−) oxidase (−)] have been serotyped with polyvalent O and Vi following manufacturer’s instructions (Denka Seiken) [24].

### 2.4. Outcome Variable

Our analyses included a 1:3 case-control ratio (NTS positive and negative) with datasets on both enrolment and follow-up HAZ, WAZ, and WHZ for the participants enrolled in GEMS. The primary measure of growth in our analyses was the length or height-for-age z-score (HAZ), weight-for-age z-score (WAZ), and weight-for-height z-score (WHZ), the average of three repeated measurements for each child at every visit was derived according to WHO guidelines [18]. GEMS removed implausible height values and values that were inconsistent between enrollment and follow-up at the time of creating the master data file for any analysis [20]. In our study, we used weighted means of baseline and endline HAZ (*n* = 344), WAZ (*n* = 346), and WHZ (*n* = 344) from enrolment to follow-up for the NTS (+) children enrolled in GEMS.

### 2.5. Variables of Interest

#### 2.5.1. Anthropometry Measurements

Height, weight, and mid-upper arm circumference (MUAC) were measured at enrollment and the 60-day follow-up visit for each child, and details of measuring methods are described elsewhere [18]. Using a digital scale calibrated every day (model 314, Tanita Corp of America, Arlington Heights, IL, USA) weight (to the nearest 0.1 kg) was recorded with the child naked or in light clothing; the weight of the mother alone and with the child was documented for children 0–23 months of age, and the weight of the child was measured during the study. In the recumbent position, the length of children 0–23 months of age or those who were older but unable to stand unassisted was measured using a board with a fixed head and sliding foot piece (to the nearest 0.1 cm) (Shorr Productions, Olney, MD, USA). In children 2 years of age and older, the same apparatus was used to measure standing height. To calculate MUAC to the nearest 0.1 cm, a 25-cm paper single-slotted insertion tape was used (Shorr Productions) [26]. Length/height and MUAC were measured three times each and the average was estimated [27]. Using the WHO Child Growth Standards as the reference population, the height/length-for-age, weight-for-age, and weight-for-height/ length z-scores (HAZ, WAZ, and WHZ) were measured using a WHO SAS macro [28,29].

#### 2.5.2. Diarrhea

Defined as the passage of ≥3 abnormally loose or watery stools per 24 h [18,30], while acute diarrhea was defined as having 1 ± 6 days of diarrhea within the last 21 days.

#### 2.5.3. Vomiting and Fever

Many of the components, such as vomiting (≥3 times/ day), fever on admission (measured at least 38 degree Celsius), and visible blood in stools can only be retrospectively assessed [18].

#### 2.5.4. Breastfeeding Status 

Breastfed referred to both exclusive and partially breastfeed children.

#### 2.5.5. Sociodemographic Information

It was about the enrolled and the child’s household (defined as a group of people sharing a common cooking fire) including mother as a primary caretaker, mother’s education (illiterate or literate), and household size (including the number of children <5 years of age and people sleeping in the house regularly). Building materials (predominant floor material: cement or noncement, e.g., made of earth, sand, dung, and others), were considered as explanatory variables. Variables addressed handwashing practices (before nursing or preparing baby food; after handling animals and cleaning a child), access and the main source of drinking water (tube well water and non-tube well water), water treatment (water treatment method of drinking water available or not), sanitation facilities (toilet facility for disposal of human fecal waste available or not), animals on the premises (sheep, goat, fowl, cow, dog, and cat), and the use of handwashing materials (water with soap or without soap).

#### 2.5.6. Wealth Index

Households were categorized based on the wealth index quintiles as socioeconomic status (SES) (poor, lower-middle, middle, upper-middle, and rich) to assess possible factors associated with disease [18,31].

#### 2.5.7. Household Follow-up Visit 

Approximately 60 days after enrollment, GEMS field workers visited each enrolled child’s household (acceptable range, 50–90 days). They measured vital status, reported in- between medical events, measured the axillary temperature of the child, and carried out anthropometric measurements, details have been described elsewhere [18]. Details of in- between comorbidity data (malaria, typhoid, pneumonia, diarrhea, and dysentery) were collected during these follow-up household visits.

### 2.6. Statistical Analysis

We reported the child, maternal, and household-level characteristics by using mean and standard deviation (SD) for continuous variables and frequency as a percentage for categorical variables to summarize the data. A paired t-test was used to assess the difference between baseline and endline (60 days’ follow-up after enrolment) z- score among these GEMS children. To assess the relationships between the explanatory variable (presence of fecal NTS) and the outcome variable (HAZ, WAZ, and WHZ), we used simple generalized estimating equations (GEE) with the Gaussian family (normal distribution) and identity link function [32]. Repeated measure was used as the time variable using *xtset* command in STATA and within-group correlation structure for population average panel data model was *Exchangeable correlation* with default standard errors. The GEE methodology is an extension of the quasi-likelihood approach that makes it possible to define a working correlation matrix at different follow-up visits for the correlation within the data obtained by repeated measurements from the same respondent over time. These outcomes are more efficient and unbiased regression parameters [33]. The explanatory variable (presence of fecal NTS) was initially used separately in the GEE model to assess its unadjusted influence on the outcome variable (WAZ, HAZ, and WHZ). For multivariable modeling, the covariates indicating the relationship with the outcome variable in the literature review were chosen. To achieve an adjusted final model, all covariates (age, gender, diarrhea at enrolment, breastfeeding status, mother’s education, number of people regularly sleep in the house, number of children under the age of 5 at the house, handwashing before nursing a child and after cleaning the child, handwashing material, main source of drinking water, wealth index, available toilet facility, co-pathogens (*Campylobacter* and *Giardia)*, comorbidity (malaria, typhoid, pneumonia, diarrhea, dysentery), and study site) were used in the multiple GEE model. To detect multicollinearity, the variance inflation factor (VIF) was calculated, and no variable producing a VIF value >5 was found in the final model. We calculated the strength of the association by estimating the coefficient and its 95% CI (confidence interval). A two-sided alpha <0.05 was considered statistically significant during the analysis. All data were analyzed using version 15.0 IC of STATA (College Station, TX, USA: Stata Corp LLC).

### 2.7. Ethical Consideration

Before the beginning of the research, the clinical procedure, consent forms, CRFs (case report forms), field procedures, and other supporting materials were approved by the ethics committees and the related scientific review boards at the University of Maryland School of Medicine and the committees overseeing each site and their coordinating partners from other institutions. The signed informed consent forms for the children’s inclusion in the study were collected from the children’s parents/guardians (both sick cases and healthy controls).

## 3. Results

### 3.1. Characteristics of the Study Population

The baseline demographic characteristics of the study participants are presented in Table 1. We enrolled a total of 1512 children (378 fecal NTS positives and 1134 NTS negative controls) from GEMS. Among them 40.2% were female, half of the children formed the 0–11 months age group. Median MUAC, HAZ, WAZ, and WHZ were 13.7 ± 1.5, −1.52 ± 1.4, −1.47 ± 1.4, and −0.87 ± 1.4, respectively, at enrollment. About 41.4% were breastfed; half of the children had moderate-to-severe diarrhea; 14.6% reported the presence of blood in the stool; and, in the case of Bangladesh site, almost half of the children reported visible blood in the stool and 50% were febrile during enrollment, while vomiting ≥3 times/day was observed less often during enrollment. About 97% of the primary caretakers were a mother and almost 70% were literate. An average of eight people regularly slept in the house, and at least one child under 5 was present at the house. Forty-three percent of families had cemented floor material. About 23% of children belonged to a middle-class family. Sixty-five percent had fowl, 44% had a cow, and 50% of families indicated the presence of dog and cat in the house. One-fourth of the households were a user of tube well water as the main source of drinking water and for domestic purposes. Around 40% used a water treatment method at the house. A toilet facility was available in 87% of the houses; in India, Mali, Mozambique, and The Gambia, 100% of respondents had a toilet facility at the house. Almost 79% used soap and water during handwashing. Handwashing practice was observed among participants around 40% of the time before nursing a child and after cleaning the child, and only 13% after handling an animal. *Campylobacter* (17%) and *Giardia* (13.6%) were reported as leading co-pathogens from the stool samples, both of which were frequently detected in the Pakistan site.

### 3.2. Child Growth and Fecal NTS

In the unadjusted model, the difference between baseline (at enrollment) and endline (on day 60 follow up, ranging from 50 to 90 days later after enrollment) z- scores of 1512 children under the age of 5 are shown in Table 2, by age group. 

The mean difference of HAZ scores was estimated between enrolment and follow-up, and significant differences were observed in the case of children 0–59 months (decreased), 0–11 months (decreased), and 12–23 months (improved). In the case of WAZ scores, a significant difference was observed in the case of children 24–59 months (improved). In the case of WHZ scores, significant differences were observed in the case of all children 0–59 months (improved), 0–11 months (improved), 12–23 months (improved), and 24–59 months (improved) (Table 2).

Among the fecal NTS-positive children under the age of five, mean HAZ score was decreased (−1.70) at the endline, but mean WAZ and WHZ scores improved in the endline (Figure 2). 

The outcomes of GEE modeling are presented in Table 3. An important correlation between fecal NTS with WAZ and WHZ (*p* < 0.05) at the endline is shown in the unadjusted GEE model. Gaussian GEE modeling, considering all variables once at a time, was conducted longitudinally. After adjusting for age, gender, diarrhea, breastfeeding status, mother’s education, number of people regularly sleeping in the house, number of children under the age of five in the house, handwashing before nursing a child and after cleaning the child, type of handwashing material, the main source of drinking water, wealth index, available toilet facility, co-pathogens (*Campylobacter* and *Giardia)*, comorbidity (malaria, typhoid, pneumonia, diarrhea, dysentery), and study site, in the multivariable model a significant negative association of WAZ score and WHZ score with fecal NTS were found (Table 3). 

## 4. Discussion

The GEMS prospective case-control study of children in seven sites made it possible to analyze the effect of the presence of fecal NTS on growth among children under the age of five. To our knowledge, this study is the first study evaluating, longitudinally, the association of fecal NTS infection with child growth.

We found a negative association of fecal NTS with weight-for-age and weight-for-height z-score, both were statistically significant. A high prevalence of severe malnutrition has been noted among young Mozambican children with invasive NTS disease [34]. Other African studies have also reported that malnutrition is a risk factor for NTS infection [15,16].

We assume that a stronger predisposing factor for NTS could be the immunocompromised condition of children with severe malnutrition. Though IL-12 (Interleukin-12) was not measured in the GEMS study and we do not know whether in our study this would have any confounding role, IL-12p40 deficiency in humans have shown an increased vulnerability to invasive NTS infection [35], which suggests that IL-12 can play a critical role in protection against NTS infection. Indeed, in malnourished children, IL-12 expression is reduced compared with well-nourished children. [36]. Another study from Bangladesh revealed severe acute malnutrition as an independent predictor for NTS bacteremia among hospitalized children [14]. We believe that the disparity between the linear growth of NTS infected children of Africa and South Asia may be due to several factors, including a low proportion of seriously immune-compromised status in South Asia, such as the absence of advanced HIV infection and burdens of diverse co-morbidities including malaria, and probable diversity in the distribution of NTS serotypes.

It is relevant since other studies have not been able to distinguish if any correlation between NTS and malnutrition have been observed from the adverse impact of non- typhoidal *Salmonella* enteritis or the increased susceptibility of malnourished children to infectious morbidity, involving a single measurement of nutritional status. 

In an animal study among *Salmonella*-infected monkeys, the mild diarrhea observed may have been the result of the colon’s inability to reabsorb the fluid load from the small intestine presented to the colon, but in some cases, reported severe diarrhea may be the result of significant abnormalities in absorption observed in both the jejunum and the ileum superimposed on the colon [37]. Mural thickening of the terminal ileum resulting in weight loss in adults was a frequently identified finding in patients with non- typhoidal *Salmonella* enterocolitis [38]. Several mechanisms may have played role in this process. Probably this may be due to reduced absorption of micronutrients in the gut and interrupted growth during childhood. Although further research is needed to know the exact pathophysiology of NTS infection and growth faltering among children under five.

In our study population, *Campylobacter* and *Giardia* were prevalent (17% and 13.6% respectively) and are also known to influence children’s growth [39,40,41]. In this case, however, their effect can also be removed, as we controlled both *Campylobacter* and *Giardia* for NTS infection during GEE modeling. Given there is a definite impact of those infections on growth, the analysis was replicated and confined to those children who were not infected with *Campylobacter* and *Giardia* (data not shown). The difference in adjusted mean growth in this small sample, whether measured in terms of the HAZ, WAZ, and WHZ score, was almost the same.

Our results have significant implications for the prevention and management of NTS in developing countries. It is widely known that children in developing countries are found to be infected with NTS but are not often suffering from diarrhea. Most of the time it causes self-limiting diarrhea and normally sick individuals are not given any antimicrobial treatment rather than supportive management. We believe that this presumption must, in the light of our observations, re-examine growth faltering/detrimental effect on the nutrition status of asymptomatic NTS-infected children.

This study also noted its limitations. It includes the inability to assess the relationship between child growth failure with maternal BMI (body mass index), gestational age, and birth weight data from GEMS. There was also a lack of data on the serum nutrient level of the enrolled children. We could not detect the effect of the human immunodeficiency virus (HIV) as HIV data in the children’s population were not readily available for South Asia. 

The strengths of our study were randomized unbiased sampling, large sample size, and high-quality laboratory performance. However, cost-free healthcare is given in both regions and is open to all individuals, irrespective of their socioeconomic or other backgrounds, raising the probability of enrollment of more people from the low socioeconomic status as study participants. Here, we looked for the association between both symptomatic and asymptomatic NTS infection and growth faltering among seven sites in children under five, which enriched the findings of the study. The single follow-up visits to the residence approximately 60 days after enrollment was a significant contribution of this study, which allowed us to explain the growth outcome of children during the vulnerable period after enrollment of at-risk children.

## 5. Conclusions

In summary, we observed the negative association between fecal NTS and weight-for-age and weight-for-height z-scores. These findings imply the need for preventive strategies targeting NTS children, which could potentially reduce the disease burden and its sequelae such as child growth faltering during the first five years of life. Accordingly, there is an urgent need for the introduction of proven effective intervention; improvements of household water, sanitation, and hygiene practice; and effective treatment of NTS gastroenteritis to reduce the challenging consequences of undernutrition in such children.

## Figures and Tables

**Figure 1 nutrients-13-00392-f001:**
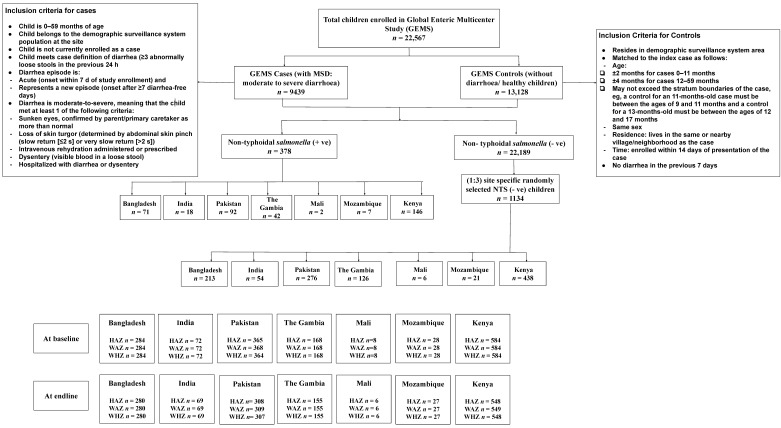
Study profile of enrolled children.

**Figure 2 nutrients-13-00392-f002:**
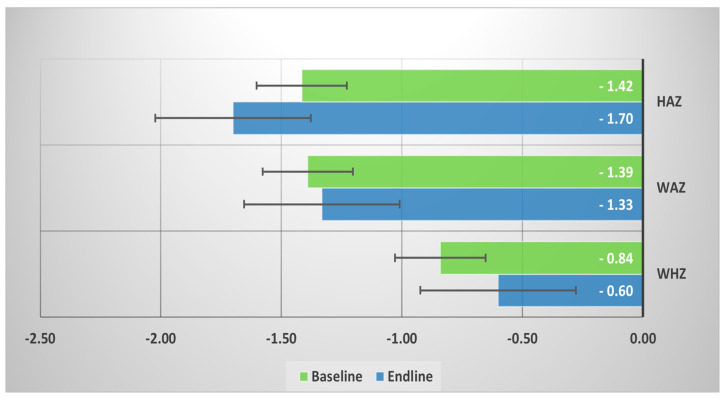
Mean baseline and endline height-for-age z-score (HAZ), weight-for-age z-score (WAZ), and weight-for-height z-score (WHZ) among the fecal NTS (+) children under 5.

**Table 1 nutrients-13-00392-t001:** Baseline characteristics of the non- typhoidal *Salmonella* (NTS) (+) children under the age of 5.

Characteristics	Bangladesh	India	Pakistan	The Gambia	Mali	Mozambique	Kenya	Overall
*n* (%)	*n* = 284	*n* = 72	*n* = 365	*n* = 168	*n* = 8	*n* = 28	*n* = 584	*n* = 1512
Non-typhoidal *Salmonella* (+)	71 (18.8)	18 (4.8)	92 (24.3)	42 (11.1)	2 (0.5)	7 (1.9)	146 (38.6)	378 (25)
Age group								
0–11 months	41 (57.8)	11(61.1)	39 (42.4)	19 (45.2)	-	6 (85.7)	68 (46.6)	184 (48.7)
12–23 months	16 (22.5)	3 (16.7)	33 (35.9)	19 (45.2)	1 (50)	1 (14.3)	46 (31.5)	119 (38.5)
24–59 months	14 (19.7)	4 (22.2)	20 (21.7)	4 (9.5)	1 (50)	-	32 (21.9)	75 (19.8)
Gender (Female)	29 (40.9)	3 (16.7)	36 (39.1)	24 (57.1)	1 (50)	2 (28.6)	57 (3.0)	152 (40.2)
Breastfed	60 (84.5)	17(94.4)	49 (53.3)	33 (78.6)	-	6 (85.7)	105 (71.9)	270 (71.4)
Baseline Anthropometry								
MUAC, median ^¶^	13.8 ± 1.3	14.2 ± 1.3	13.1 ± 1.4	13.8 ± 1.4	14.6 ± 2.4	13.2 ± 1.5	14.0 ± 1.6	13.7 ± 1.5
HAZ ^¶^	−1.09 ± 1.1	−1.09 ± 1.2	−2.18 ± 1.4	−1.41 ± 1.8	−1.47 ± 2.4	−1.95 ± 1.6	−1.39 ± 1.4	−1.52 ± 1.4
WAZ ^¶^	−1.32 ± 1.3	−1.05 ± 1.4	−2.15 ± 1.3	−1.36 ± 1.5	−1.87 ± 2.0	−1.87 ± 1.5	−1.18 ± 1.4	−1.47 ± 1.4
WHZ ^¶^	−0.95 ± 1.3	−0.63 ± 1.2	−1.33 ± 1.3	−0.77 ± 1.8	−1.69 ± 1.3	−0.93 ± 1.0	−0.58 ± 1.4	−0.87 ± 1.4
Clinical features								
Diarrhea	42 (59.2)	4 (22.2)	39 (42.4)	17 (40.5)	2 (100)	5 (71.4)	81(55.5)	190 (50.3)
Visible blood in stool	34 (47.9)	1 (5.6)	9 (9.9)	2 (4.8)	-	1 (14.3)	8 (5.5)	55 (14.6)
Fever	33 (46.5)	7 (38.9)	34 (36.9)	9 (21.4)	1 (50)	3 (42.7)	103(70.6)	190 (50.3)
Vomiting ≥3 times/day	6 (8.5)	3 (16.7)	17 (18.5)	10 (23.8)	1 (50)	-	47 (32.2)	84 (22.2)
Socio-demographic characteristics								
Primary caretaker mother	71 (100)	18 (100)	88 (95.7)	40 (95.2)	2 (100)	7 (100)	141 (96.6)	367 (97.1)
Mother’s education (literate)	66 (92.9)	14 (77.8)	30 (32.6)	38 (90.5)	1 (50)	7 (100)	144 (98.6)	266 (70.4)
People regularly sleep in the house ^¶^	5.3 ± 2.1	4.9 ± 2.1	8.7 ± 4.8	24.7 ± 18.3	16.5 ± 0.7	5.8 ± 2.9	4.7 ± 1.7	8.1 ±9.1
Under 5 children in the house ^¶^	0.7 ± 0.6	0.3 ± 0.6	0.9 ± 1.3	1.9 ± 2.9	1.5 ± 0.7	0.9 ± 0.7	0.9 ± 0.9	0.9 ±1.4
Predominant floor (cement)	14 (19.7)	17 (94.4)	63 (68.5)	29 (69.1)	2 (100)	7 (100)	31 (21.2)	163 (43.1)
Wealth index								
Poor	14 (19.7)	5 (27.8)	10 (10.9)	12 (28.6)	-	1 (14.3)	20 (13.7)	62 (16.4)
Lower middle	13 (18.3)	3 (16.7)	25 (27.2)	8 (19.1)	-	2 (28.6)	33 (22.6)	84 (22.2)
Middle	16 (22.5)	3 (16.7)	24 (26.1)	7 (16.7)	-	2 (28.6)	37 (25.3)	89 (23.5)
Upper middle	14 (19.7)	5 (27.8)	15 (16.3)	9 (21.4)	1 (50)	1 (14.3)	18 (12.3)	63 (16.7)
Richest	14 (19.7)	2 (11.1)	18 (19.6)	6 (14.3)	1 (50)	1 (14.3)	38 (26.0)	80 (21.2)
Animal present at the house								
Sheep	2 (2.8)	-	-	31 (73.8)	1 (50)	-	46 (31.5)	80 (21.2)
Goat	11 (15.5)	7 (38.9)	8 (8.7)	30 (71.4)	-	1 (14.3)	88 (60.3)	145 (38.4)
Cow	39 (54.9)	1 (5.6)	4 (4.4)	16 (38.1)	-	-	107 (23.3)	167 (44.2)
Fowl	41 (57.8)	9 (50)	13 (14.1)	38 (90.5)	-	4 (57.1)	141 (96.6)	246 (65.1)
Cat	63 (88.7)	17 (94.4)	8 (8.7)	7 (16.7)	-	-	101 (69.2)	196 (51.9)
Dog	60 (84.5)	18 (100)	3 (3.3)	13 (30.9)	-	-	103 (70.6)	197 (52.1)
WASH								
Main source of drinking water (tube well)	68 (95.8)	-	-	8 (19.0)	-	-	5 (3.4)	81(21.4)
Use a water treatment method	1 (1.5)	11 (61.1)	31 (36.1)	13 (36.1)	-	-	88 (61.9)	144 (40.1)
Toilet facility available in the house	65 (91.6)	18 (100)	89 (96.7)	42 (100)	2 (100)	7 (100)	107 (73.3)	330 (87.3)
Handwash with water and soap	52 (73.2)	7 (38.9)	60 (65.2)	32 (76.2)	2 (100)	6 (85.7)	139 (95.2)	298 (78.8)
Practice handwashing								
Before nursing a child	22 (30.9)	14 (77.8)	32 (34.8)	14 (33.3)	1 (50)	5 (71.4)	40 (27.4)	128 (33.9)
After handling animal	22 (30.9)	-	8 (8.7)	8 (19.1)	-	-	11 (7.5)	49 (12.9)
After cleaning the child	22 (30.9)	15 (83.3)	54 (58.7)	36 (85.7)	-	1 (14.3)	38 (26.0)	166 (43.9)
Toilet facility available	65 (91.6)	18 (100)	89 (96.7)	42 (100)	2 (100)	7 (100)	107 (73.3)	330 (87.3)
Co pathogens isolated in stool								
*Campylobacter*	16 (22.5)	1 (5.6)	32 (34.8)	1 (2.4)	-	-	15 (10.3)	65 (17.2)
*Giardia*	5 (7.0)	4 (22.2)	18 (19.6)	3 (7.1)	1 (50)	1 (14.3)	19 (13.0)	51 (13.6)

Abbreviation: ^¶^ Mean ± standard deviation; height-for-age z-score: HAZ, weight-for-age z-score: WAZ, weight-for-height z-score: WHZ, MUAC: median (mean; for children <5 years of age) mid-upper arm circumference, diarrhea: 3 or more stool/day, fever: measured at least 38 degree Celsius. NTS, non- typhoidal *Salmonella*, WASH: water, sanitation, and hygiene.

**Table 2 nutrients-13-00392-t002:** Difference between baseline and endline height-for-age z-score, weight-for age z-score, and weight-for-height z-score among the children under the age of 5 (*n* = 1512) of Global Enteric Multicenter Study (GEMS), by age group.

Z- Score	Age Group	Mean at Baseline (95% CI)	Mean at Endline (95% CI)	* Mean Diff (95% CI)	*p* Value
HAZ	Overall	−1.44 (−1.51, −1.38)	−1.61 (−1.68, −1.55)	−0.17 (0.14, 0.19)	<0.001
	0–11 m	−1.11 (−1.21, −1.01)	−1.36 (−1.46, −1.26)	−0.25 (−0.30, −0.20)	<0.001
	12–23 m	−1.61 (−1.73, −1.49)	−1.79 (−1.91, −1.67)	0.18 (−0.22, −0.14)	<0.001
	24–59 m	−1.77 (−1.90, −1.65)	−1.79 (−1.92, −1.67)	−0.02 (−0.06, 0.02)	0.359
WAZ	Overall	−1.26 (−1.33, −1.19)	−1.23 (−1.29, −1.16)	0.03 (−0.07, 0.01)	0.131
	0–11 m	−1.09 (−1.21, −0.98)	−1.12 ( −1.23, −1.01)	−0.02 (−0.09, 0.04)	0.458
	12–23 m	−1.33 (−1.45, −1.20)	−1.32 (−1.44, −1.19)	0.01 (0.05, 0.07)	0.788
	24–59 m	−1.43 (−1.55, −1.31)	−1.28 (−1.40, −1.17)	0.14 (0.08, 0.21)	<0.001
WHZ	Overall	−0.63 (−0.69, −0.55)	−0.49 (−0.56, −0.43)	0.13 (0.08, 0.19)	<0.001
	0–11 m	−0.51 (−0.62, −0.39)	−1.45 (−0.56, −0.33)	0.60 (−0.03, 0.16)	<0.001
	12–23 m	−0.77 (−0.89, −0.65)	−0.63 (0.75, 0.51)	0.14 (0.06, 0.23)	<0.001
	24–59 m	−0.64 (−0.77, −0.51)	−0.39 (−0.51, −0.28)	0.24 (0.14, 0.35)	<0.001

Abbreviation: HAZ: Height for age z- score, WAZ: Weight for age z- score, WHZ: Weight for height z- score; SD: Standard deviation; CI: Confidence interval; * Diff = endline-baseline; Baseline: during enrollment, Endline: on Day 60 follow up (follow-up 50–90 days later).

**Table 3 nutrients-13-00392-t003:** Association of non-typhoidal *Salmonella* (NTS) infection with child’s height-for-age, weight-for-age, and weight-for-height z-scores: results of generalized estimating equations modelling (dependent variables—height-for-age, weight-for-age, and weight-for-height z-scores).

Z-Score	Non-Typhoidal *Salmonella*			
	Unadjusted		Adjusted *	
	Coef. (95% CI)	*p* Value	Coef. (95% CI)	*p* Value
Height-age z-score	−0.90 (−0.24, 0.05)	0.216	−0.13 (−0.27, 0.01)	0.073
Weight-age z-score	−0.18 (−0.33, −0.03)	0.021	−0.19 (−0.33, −0.04)	0.010
Weight-height z-score	−0.19 (−0.34, −0.05)	0.010	−0.19 (−0.34, −0.04)	0.007

* adjusted for age, gender, diarrhea, breastfeeding status, mother’s education, number of people regularly sleep in the house, number of children under the age of 5 in the house, handwashing before nursing a child and after cleaning the child, handwashing material, main source of drinking water, available toilet facility, wealth index, co-pathogens (*Campylobacter* and *Giardia*), site, and comorbidity (malaria, typhoid, pneumonia, diarrhea, dysentery). Abbreviation: Coef.: coefficient, CI: confidence interval.

## Data Availability

A publicly available GEMS dataset was analyzed in this study. This data can be obtained from here: ClinEpiDB [https://clinepidb.org/ce/app/record/dataset/DS_841a9f5259].

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
