# Peer review of "Association between Non-Typhoidal Salmonella Infection and Growth in Children under 5 Years of Age: Analyzing Data from the Global Enteric Multicenter Study"

_nutrients, 2021, doi:10.3390/nu13020392_

Round 1

Reviewer 1 Report

The authors describe a secondary analysis of the Global Enteric Multi-Center Study (GEMS) data to investigate the association between non-typhoidal Salmonella infection with growth in children under 5 years of age. The manuscript is generally well-written and describes and important analysis of an understudied area using these publicly available data. However, I have some concerns regarding the analytic approach as described. My specific comments are provided below.

Abstract:

  • Minor – The statement at line 27 that “The association between linear growth and fecal NTS (p > 0.05) was not statistically significant” could be improved by providing the actual p-value and a measure of the effect size space permitting. Providing the p-value alone does not give much information in the absence of a clearly stated decision boundary and sample size justification (which are often not directly relevant in such observational research).

  • Minor – The sentence “In the GEMS, where children were followed during 50-90 days of enrolment, the presence of fecal NTS harmed the child’s nutritional status” seems a bit vague given the focus here was on child anthropometrics specifically. I would recommend replacing the term “child nutritional status” with a statement specific to the anthropometric outcomes examined.

Introduction:

  • Minor: The first sentence is not clear. “An approximate 3.4 million global cases per year are caused by non-typhoidal Salmonella (NTS) infection”. Global cases of what? I think this just needs rewording.

Methods:

  • Major – It is unclear from figure 1 and the text how the authors came to 1,134 non-Salmonella exposures participants for this analysis. In the parent study, were samples from only 378 + 1,134 processed/analyzed/assayed for NTS? Or were samples from all 22,567 processed for NTS? If all were processed, and the goal was increase statistical power, then why not include all the NTS- participants in the analysis as long as their covariate distribution was similar to the NTS+? It is also not clear as written whether the matching was individual or frequency in the parent study (reads as if perhaps individual matching) and why the original matches were broken for this analysis if that was the case (more detail needed to understand the implications of this decision). 

  • Minor – “GEMS removed implausible height values and values that were inconsistent between enrollment and follow-up at the time of creating the master data file for any analysis”. How was this defined/operationalized?

  • Major – The statement, “We used weighted means of baseline and endline HAZ (n= 344), WAZ (n= 346), and WHZ (n= 344) from enrolment to follow-up for the NTS (+) children” is unclear given this is described as a longitudinal analysis/study. In the abstract it is stated that association between an explanatory variable (NST) and the outcome variable (WLZ) was longitudinally tested using generalized estimating equations; however, here it reads as if the outcome measures were perhaps somehow averaged prior to the analysis…and maybe only for the NTS+ children? Greater clarity is needed here.

  • Major – Similar to the point above, greater clarity is required regarding the approach used to model the association between NTS and the anthropometric outcomes. For the GEE model, the distribution and link function, cluster term, working covariance structure, and fixed effect terms should be described in detail (as well as the how the standard errors were obtained if there are different options for this is STATA). In particular, this reads as if time, or the time-by-NTS interaction, were not included as model terms. If this is the case, then the authors report marginal estimates for the outcomes taken over all time points and subjects; not a longitudinal analysis of these data. Typically, time might be included to estimate a time-averaged model and the time-by-NTS used to obtain estimates for the change over time (as one example of how this could be fit).  In addition, there is no description of how missing data were handled (GEE models make the rather strong assumption that any data are missing completely at random) and no discussion of how site was accounted for in these models. Without this additional detail, I do not know who to interpret the results that are presented. 

  • Minor – Did the authors consider if the associations differed according to site (say S. Asia vs. Africa) or according to participant age or other characteristics of possible interest? Would this information be of substantive interest?

  • Minor – I suspect the sentence “Data were visualized using various types of plots such as box plots, histograms, pie charts, bar diagrams, scatter plots, etc.” could be omitted if these results are not presented in the manuscript.

  • Major – The authors state, “To detect multicollinearity, the variance inflation factor (VIF) was calculated, and variables producing a VIF value > 5 were omitted from the study to remove collinearity”. Which variables were considered and removed if they are not among the final model terms presented?

  • Minor – This is a little pedantic, but “A probability of less than 0.05 was considered statistically significant during the analysis.” Is not complete as it does not describe the probability of what. More common one might state that a two-sided alpha < 0.05 was …

Reviewer 2 Report

The paper presented to me for review presents an analysis of the relationship between non-typhoidal Salmonella infection and childhood development. The topic of the work is very important for epidemiological, economic and social reasons. As stated in the paper, Salmonella infections still cause a very high number of patients, especially in countries with low economic development. The study analysed data from 7 countries in the Global Enteric Multicentre Study. The study was conducted over 4 years. In the presented paper the authors analysed more than 22 thousand described cases. In the prepared publication the authors presented in detail the statistical methods applied in this type of work. In Table 1 they presented the obtained research results in a very detailed way. In my opinion the authors should consider presenting the research results in a more simplified and transparent way. I have no major objections to the remaining parts of the publication. Therefore, before accepting the paper for publication, I recommend improving the presentation of the results presented in Table 1.

In my opinion, the introduction should also briefly discuss the topic of chronic salmonella infections.

Round 2

Reviewer 1 Report

NA